# Transcriptome Revealed Exposure to the Environmental Ammonia Induced Oxidative Stress and Inflammatory Injury in Spleen of Fattening Pigs

**DOI:** 10.3390/ani12091204

**Published:** 2022-05-07

**Authors:** Yongjie Chen, Runxiang Zhang, Susu Ding, Haoyang Nian, Xiangyin Zeng, Honggui Liu, Houjuan Xing, Jianhong Li, Jun Bao, Xiang Li

**Affiliations:** 1College of Animal Science and Technology, Northeast Agricultural University, Harbin 150030, China; chenyongjie97@163.com (Y.C.); zhangrunxiang@neau.edu.cn (R.Z.); dingssu@163.com (S.D.); 18846076532@163.com (H.N.); xiangyin_1996@163.com (X.Z.); liuhonggui1312@163.com (H.L.); xinghoujuan@neau.edu.cn (H.X.); ryan626@139.com (X.L.); 2Key Laboratory of Swine Facilities, Ministry of Agriculture, Northeast Agricultural University, Changjiang Road No. 600, Harbin 150030, China; 3College of Life Science, Northeast Agricultural University, Harbin 150030, China; jhli@neau.edu.cn

**Keywords:** fattening pigs, ammonia, transcriptomics, spleen, oxidative stress, inflammatory

## Abstract

**Simple Summary:**

Ammonia is a major environmental pollutant. Previous estimates of ammonia emissions have focused on livestock sources in agricultural systems. Livestock continues to be the main source of ammonia emissions. Exposure to high concentrations of ammonia can cause varying degrees of damage to tissues and organs. However, the damage of ammonia exposure to the spleen of pigs in the fattening pigs is unknown. Therefore, the aim of this study was to explore the mechanism at the gene level of exogenous ammonia-induced spleen toxicity by enzyme-linked immunosorbent assay (ELISA), spleen histomorphological observation, and transcriptome technology. The results showed that ammonia exposure led to oxidative stress, activation of inflammatory pathways, and splenic injury. In addition, the genes that encode histone methyltransferase were found to be significantly upregulated. Therefore, histone methylation may be the epigenetic mechanism of splenic poisoning induced by ammonia. Our findings provide a novel direction for exploring the underlying molecular mechanisms of ammonia toxicity.

**Abstract:**

Ammonia is one of the major environmental pollutants that seriously threaten human health. Although many studies have shown that ammonia causes oxidative stress and inflammation in spleen tissue, the mechanism of action is still unclear. In this study, the ammonia poisoning model of fattening pigs was successfully established. We examined the morphological changes and antioxidant functions of fattening pig spleen after 30-day exposure to ammonia. Effects of ammonia in the fattening pig spleen were analyzed from the perspective of oxidative stress, inflammation, and histone methylation via transcriptome sequencing technology (RNA-seq) and real-time quantitative PCR validation (qRT-PCR). We obtained 340 differential expression genes (DEGs) by RNA-seq. Compared with the control group, 244 genes were significantly upregulated, and 96 genes were significantly downregulated in the ammonia gas group. Some genes in Gene Ontology (GO) terms were verified and showed significant differences by qRT-PCR. The KEGG pathway revealed significant changes in the MAPK signaling pathway, which is strongly associated with inflammatory injury. To sum up, the results indicated that ammonia induces oxidative stress in pig spleen, activates the MAPK signaling pathway, and causes spleen necrosis and injury. In addition, some differential genes encoding epigenetic factors were found, which may be involved in the response mechanism of spleen tissue oxidative damage. The present study provides a transcriptome database of ammonia-induced spleen poisoning, providing a reference for risk assessment and comparative medicine of ammonia.

## 1. Introduction

Ammonia is a toxic, colorless, and corrosive alkaline gas, which is recognized as an atmospheric pollutant [1]. Sources of ammonia include industrial production, automotive exhaust, agriculture, livestock, construction, and soil and marine volatilization [2]. It is noteworthy that the ammonia emission from husbandry accounts for an important share of global ammonia emissions [3]. The excessive ammonia in livestock and poultry houses has an adverse effect on the health of workers and the welfare of animals, causing air pollution through the ventilation system. Therefore, the harm of ammonia emission from the livestock and poultry industry to humans and animals has become a hot issue in the world [4].

Inhalation is the primary route of exposure to ammonia. Ammonia below 25 ppm is harmless to both humans and animals [5]. However, exposure to high concentrations of ammonia in humans can cause the destruction of respiratory tissues, cilia, and the mucosal barrier, leading to inflammation and an increased risk of secondary infections [6]. The spleen is the largest peripheral immune organ, more complex than other lymphoid tissues, and it is crucial to the maintenance of innate immunity and acquired immune function [7]. A recent study shows that ammonia can stimulate an immune response in pufferfish and promote the production of various inflammatory cytokines [8]. However, the mechanism of ammonia-induced splenic injury remains unclear.

Epigenetic modifications are responses to changes in the environment after environmental stimuli (nutrients, toxins, infections, hypoxia) [9]. Histone modification is an important epigenetic modification. Post-translational methylation of histones has been extensively studied in the past few years. Mitogen-activated protein kinase (MAPKs) regulate multiple life processes [10]. P38 MAPK and JNK/SAPK are mainly induced by various environmental stresses. PM2.5 exposure was found to activate MAPK signaling pathways, and phosphorylated expressions of JNK, P38, and ERK were significantly increased [11]. The nuclear factor kappa-B (NF-κ B) signaling network is one of the histone modification pathways that regulate the expression of inflammatory factors. It has been shown to be a downstream consequence of MAPK signaling [12].

Transcriptomes are essential for explaining the function of the genome, the molecular makeup of cells and tissues, and the onset and development of disease [13]. In recent years, pigs have been widely studied in the field of toxicology as a traditional animal model [14] because the life and metabolic system of pigs is similar to that of humans, and pigs allow for repeated sampling [15]. Ammonia is the major noxious gas in pig farms. The impacts of ammonia on livestock and health have been studied relatively well in chickens. However, less research has been conducted on the mechanism of ammonia toxicity to pig spleen.

To study how exogenous ammonia causes spleen damage, we established an ammonia poisoning model of fattening pigs to conduct spleen ultrastructural examination. Simultaneously, we used RNA-seq to analyze the impact of ammonia on spleen tissue and to quantify the variation of expression levels of transcripts under different conditions. We investigated the oxidative stress and inflammatory injury induced by ammonia in the spleen. These research results may provide new ideas and insights for understanding the specific mechanisms of ammonia toxicity.

## 2. Materials and Methods

### 2.1. Experimental Animal Management and Experimental Design

Twelve 125-day-old crossbred pigs with similar body conditions were randomly chosen and indiscriminately distributed into control group (*n* = 6) and ammonia group (*n* = 6). All experimental animals were housed in an ambient temperature-controlled room. All environmental parameters were in accordance with the national standard for fattening pig breeding (GB17824.3-2008). All pigs made use of feed and water freely.

### 2.2. Ammonia Exposure

For the purpose of simulating the pig houses and ensuring that the ammonia concentration of the control group was maintained below 5 mg/m^3^ and the ammonia concentration of the ammonia group was maintained between 75.4 and 76.5 mg/m^3^, the exogenous ammonia was supplied to the ambient temperature-controlled room for a time of 8 h/d. The pigs were adapted to the feeding environment 14 d before the formal experiment, and the ammonia concentration was adjusted 3 d prior to the experiment. The test period lasted for 30 d. After the experiment, the pigs were euthanized, and spleen tissues were taken immediately after blood collection and homogenized under ice conditions. The spleen tissues were taken, washed with sterile saline, and fixed in 10% formalin. Tissue homogenates were carried out for the detection of antioxidant indices, and the rest of the spleen tissues were transferred into DNase/RNase-free cryopreservation tubes and stored at −80 °C for transcriptome and qRT-PCR analysis.

### 2.3. Histological Observation

The spleen tissues were fixed in 10% formalin for at least 24 h and then pruned, cleaned with tap water, dehydrated at different concentrations of alcohol, and embedded in paraffin. The samples were cut into 6 μm thin sections and stained with hematoxylin and eosin (H&E). Finally, the stained slices were sealed with neutral resin. The changes in microstructure were recorded with a multifunctional digital light mirror DMS-651 observation.

### 2.4. Determination of Antioxidant Indexes by ELISA

The spleen tissues were homogenized in 0.9% NaCl solution and centrifuged at 3500 rpm for 10 min at 4 °C. The supernatant was taken, and its antioxidant activity was determined with a specialized assay kit. The contents of glutathione (GSH) (Kit Number: A006-2-1) and malondialdehyde (MDA) (Kit Number: A003-1) and the activity of glutathione peroxidase (GSH-px) (Kit Number: A005) and superoxide dismutase (SOD) (Kit Number: A001-1) in spleen tissue were tested by a spectrophotometric microplate reader (SpectraMax^®^ ABS00254, Molecular Devices, San Jose, CA, USA). Optical density (OD) values at 405 nm, 532 nm, 412 nm, and 550 nm were measured correspondingly. The quantitative analysis of total protein in the spleen tissue was performed by the Coomassie blue method. All kits were purchased from Nanjing Jiancheng Institute of Biological Engineering (Nanjing Construction Bioengineering Research, Nanjing, China).

### 2.5. RNA-seq and Interpretation of Results

The total RNA was extracted with TRIzol reagent. We purified the total RNA to obtain the mRNA and then decomposed it into small fragments 200–300 bp long. The first-strand cDNA was synthesized as a random hexamer, and then the second-strand cDNA was synthesized by adding buffer, dNTP, RNase H, and DNA polymerase I. Finally, PCR amplification was performed to obtain the final sequencing library. Sequencing data were screened and sorted by the Illumina HiseqTM 6000 sequencer, and the sequencing read length was double-ended 2 * 150 bp (PE150). The entire transcriptome analysis procedure was performed by Lianchuan Biotechnology Co., Ltd. (Hangzhou, China). The study of DEGs was carried out based on the final sequencing library (threshold: |log2foldchange| ≥1, *p* < 0.05). EdgeR was used for differential expression analysis. The GO database was downloaded for DEG ontology analysis. The KEGG database was used for gene function annotation. The R language was used for a graphical display of the differential expression results, including a heat map, scatter map, and volcano map of DEGs.

### 2.6. Verification of Differentially Expressed Genes

A 0.1 g amount of cryopreserved spleen tissue was weighed. Trizol (Invitrogen, Shanghai, China) was applied to extract total RNA. Firstly, 1 mL of TRIzol was added to the tissue, the homogenate of the tissue was taken out, and 200 μL of chloroform was added for shaking and standing centrifugation. Then, 400 μL of isopropyl alcohol was added to the supernatant. The precipitation was washed with 1 mL of 75% ethanol, and the dried RNA was gently blown with 30 μL of double-distilled water to dissolve. The complementary DNA (cDNA) was synthesized using oligo dT primers and reverse transcriptase II. The cDNA was diluted 5 times in double-distilled water and stored at −80 °C for later use.

Primer Premier 5.0 was used to design specific primers. qRT-PCR was performed using LightCycler^®^96 (Roche, Switzerland). The reaction conditions were as follows: three replicates were performed for each PCR product. The relative mRNA expression levels of each gene were calculated by 2^−ΔΔCt^. To corroborate the reliability of the DEGs data, we selected 13 genes associated with oxidative stress, inflammation, histone methylation, and the MAPK signal pathway and used qRT-PCR for investigation. Primers used for qRT-PCR were displayed in Table 1. β-actin was used as a reference gene to analyze the expression trend of the screened genes.

### 2.7. Statistical Analysis

IBM SPSS statistical software (version 25.0, SPSS Inc., Chicago, IL, USA) was used to analyze the experimental data. The data are expressed as mean ± standard deviation. One-way analysis of variance confirmed a significant difference between the ammonia and control groups. *p* < 0.05 was considered statistically significant.

## 3. Results

### 3.1. Splenic Ultrastructural Damage due to Ammonia Exposure

To more intuitively demonstrate the effect of ammonia on the spleen, we evaluated the spleen pathology in the control group and ammonia group of fattening pigs. The changes in the microstructure of pig spleen caused by inhaling ammonia are shown in Figure 1. In the normal spleen, the division of red pulp and white pulp was clear, the shape of splenic nodules was regular, and the number of lymphocytes was abundant. Histopathologically, the size of splenic nodules and the number of lymphocytes were decreased in the ammonia group. Pathological results indicated that ammonia exposure caused structural changes in spleen tissue. The changes in the number of lymphocytes suggested the inflammatory injury in spleen tissue.

### 3.2. Effect of Excessive Ammonia on Oxidative Stress Indexes

Oxidative stress could induce the activation of inflammatory signaling cascades. In order to investigate the mechanism of splenic pathological injury caused by excessive ammonia, we detected the activity of GSH-Px and SOD and the content of MDA and GSH. As shown in Figure 2, the GSH content of the ammonia group and the activity of GSH-Px and SOD were significantly reduced (*p* < 0.05) compared to the control group. Conversely, the MDA content of the ammonia group increased significantly (*p* < 0.05). Collectively, these data indicated that spleen injury may be induced by the oxidative stress pathway in ammonia groups.

### 3.3. Differentially Expressed Gene Screening Analysis

Our preliminary results suggested that ammonia exposure causes pathological damage and oxidative stress in the spleen. To further explore the molecular mechanism of spleen injury, we compared gene expression levels between the two groups via transcriptome sequencing. Ammonia-induced gene expression changes are shown in Figure 3. A total of 340 differential genes were screened according to log2 FC ≥1 or ≤−1 and *p* < 0.05, of which 244 genes were significantly increased, and 96 genes were significantly decreased. These results suggested that ammonia poisoning has a complex molecular regulation mechanism, which requires further analysis and clarification.

### 3.4. Ammonia Exposure Activated MAPK Signaling Pathway by KEGG Enrichment Analysis

The MAPK signaling pathway is a classic inflammatory signaling pathway. Excitedly, we obtained significant changes in the MAPK signaling pathway by KEGG enrichment analysis. The enrichment analysis results were represented by scatter plots (Figure 4). A total of 20 KEGG pathways were obtained, and these pathways pertained to five branches. As shown in Figure 4, DEGs were concentrated in the MAPK signaling pathway. A total of 12 DEGs were enriched in this pathway. It was noteworthy that this pathway was connected with oxidative stress and inflammation.

### 3.5. GO Enrichment of DEGs

GO contains three ontologies, which can be described as the biological process, cellular component, and molecular function of genes. The histogram of GO enrichment analysis results reflected the number distribution of differential genes on the three ontologies. DEGs were enriched in 307, 44, and 106 items in biological processes, cell components, and molecular functions, respectively. Among them, the biological process with the highest enrichment degree of differential genes was regulation of transcription and DNA templates, cell components were membrane, and molecular functions were protein binding. We selected the top 20 GO terms for display. A rich factor indicates the degree of GO enrichment. The results of the enrichment bubble diagram are shown in Figure 5B. Among them, histone H3-K4 methylation was the most affected.

### 3.6. qRT-PCR Confirmed the Availability of Transcriptome Data

Histone methylation is an epigenetic mechanism by which cells respond to environmental stimuli. Combined with our previous research results, we speculated that the mechanism of ammonia poisoning might be related to histone methylation. To confirm the reliability of transcriptomic results, we performed qRT-PCR on 13 genes associated with histone methylation (KMT2A, KMT2B, KMT2C, KMT2D), inflammation (IL1RL1, IL17RB, IL23R), oxidative stress (UPF1, TRPM6, AOC2), and the MAPK signaling pathway (PRKCG, CACNA2D2, HGF). As shown in Figure 6, with the exception of CACNA2D2, mRNA levels were significantly higher. Transcriptome results showed that KMT2A, KMT2B, KMT2C, KMT2D, IL1RL1, IL17RB, IL23R, UPF1, TRPM6, AOC2, PRKCG, and HGF were upregulated DEGs. CACNA2D2 was downregulated DEGs. Changes in these genes detected by qRT-PCR confirmed the credibility of the transcriptome sequencing results.

## 4. Discussion

The spleen is an important immune lymphoid organ in animals with a variety of immune functions [16]. Many studies have shown that exogenous ammonia exposure causes damage to a variety of tissues and organs, including the spleen [17,18]. In the current study, we analyzed the morphological damage and oxidative damage levels of the spleen in fattening pigs exposed to ammonia. Moreover, we used RNA-seq and qRT-PCR (KMT2A, KTM2B, KMT2C, KMT2D, IL1RL1, IL17RB, IL23R, UPF1, TRPM6, AOC2, PRKCG, CACNA2D2, HGF) to construct the database of DEGs. Transcriptome profiling revealed that the inflammation and oxidative stress caused by ammonia exposure may be related to histone methylation modification.

An imbalance between reactive oxygen species (ROS) and antioxidant mechanisms leads to oxidative stress [19]. However, exposure to harmful exogenous factors will generate ROS. High concentrations of ROS can adversely modify cellular components such as lipids, proteins, and DNA [20]. Mammalian plasma contains quite active AOC, namely plasma or SAO. As a byproduct of amine oxidation, AOC3 could directly increase reactive oxygen species levels [21]. In this study, AOC2 expression levels increased remarkably. Therefore, we hypothesized that the upregulation of the AOC2 gene in ammonia-induced spleen suggests that its function may be related to oxidative stress. This may provide an important reference for studying the function of the AOC2 gene. UPF1 has been initially identified as a central component of the NMD pathway [22]. It could regulate the expression of related genes through hnRNP E2 to affect ROS production and further induce oxidative stress [23]. The intracellular redox states can significantly alter the gated properties of ion channels. In oxidative stress, TRPM6 could restore its activity by methionine sulfoxide reductase [24]. MDA is the principal and most studied product of polyunsaturated fatty acid peroxidation as a marker of ROS. The measurements of GSH, SOD, and GSH-px could indicate antioxidant responses [25]. In this study, ammonia exposure significantly affected the activities of GSH, GSH-px, SOD, and MDA in spleen tissues. The results demonstrated that ammonia damaged the antioxidant defense system of pigs.

Excessive production of ROS in the mitochondria causes oxidative damage and activates inflammatory signaling cascades [26]. Ammonia exposure has been shown to trigger inflammatory responses and oxidative stress [27]. The current findings were in agreement with many studies. The KEGG pathways obtained by RNA-seq were related to inflammation, such as inflammatory mediator regulation of TRP channels, the MAPK signaling pathway, and the IL−17 signaling pathway. MAPK signaling has been confirmed to mediate inflammatory responses [28]. Previous reports have shown that ROS accumulation activates the NF-κ B/MAPK signaling pathway and increases downstream levels of associated inflammatory cytokines. The increase in ROS in cells activates ERKs, JNKs, or P38 MAPKs, but the mechanism by which ROS activates these kinases is unclear [29]. The PRKCG gene encodes PKC γ. PKC γ directly activates members of the MAPKs family involved in pain and multiple-injury-activated pathways [30]. The CACNA2D2 gene encodes a functionally auxiliary subunit of voltage-gated Ca^2+^ channels. Previous studies have found the reduced activation of MAPK cascade signaling in CACNA1C deficient mice [31]. HGF is a cytokine with strong angiogenic activity, which can activate the MAPK signaling pathway and upregulate the expression of iNOS [32]. The results of this study showed that the gene expression levels of PRKCG, CACNA2D2, and HGF were significantly changed in the spleen of pigs exposed to ammonia, suggesting that ammonia exposure activated the MAPK signaling pathway. Consistent with our results, Wang et al. reported that Pb exposure induces necrotic apoptosis of the chicken spleen by activating the MAPK/NF-κ B pathway [33]. The MAPK signaling pathway activates and produces a large number of cytokines that are essential for the functioning of the inflammatory response [34]. Our transcriptome results suggest that IL1RL1, IL17RB, and IL23R pertained to the active regulation of inflammatory responses. IL1RL1, a receptor for interleukin-33 (IL-33), plays a crucial part in type 2 inflammation [35]. Il-17 is an inflammation-promoting cytokine that is produced primarily by activated T cells. It enhances the activation of T cells and stimulates a variety of cells to produce inflammatory mediators, leading to the induction of inflammation [36]. IL23R is expressed in multiple cell types. It has been found to play a key role in cytokine signal transduction and amplification mediated by human macrophages [37]. Histopathological analysis further demonstrated that ammonia could cause inflammatory injury of the spleen in fat pigs.

Histone methylation is a crucial type of epigenetic modification in cells [38]. The effects of oxidative stress on chromatin modification mediate many cellular changes, including regulation of gene expression, apoptosis, and mutations [39]. Histone methylation and acetylation changed significantly after ROS exposure in human bladder tissues [40], which then lead to aberrant gene expression, and may contribute to the process of carcinogenesis. In the present study, GO analysis emphasized that some genes were gathered in “histone methyltransferase activity”. KMT2A, KMT2B, KMT2C, and KTM2D are members of the mammalian H3-K4 methyltransferases family, which regulate gene expression by catalyzing methylation on lysine 4 of histone H3 (H3K4) [41]. Transient receptor potential melastatin IDH2 is the major NADPH-producing enzyme in mitochondria. CDH1 ubiquitinates IDH2 and promotes the increased ROS in mitochondria [42]. Transcriptome results showed that the histone lysine methyltransferase 2 (KMT2) gene family changed significantly. Our qRT-PCR results showed that oxidative stress and inflammatory response induced by ammonia exposure were related to histone methylation modification.

## 5. Conclusions

In conclusion, the transcriptome map of differential gene expression in the spleen induced by ammonia was established in the present study. Our results suggest that exposure to exogenous ammonia can induce oxidative stress and inflammatory response in spleen tissues. In addition, our study found differences in histone methylase gene expression caused by ammonia exposure for the first time, which may be the epigenetic mechanism of ammonia poisoning of the spleen. The results of this paper provide a new perspective for exploring the mechanism of spleen poisoning induced by ammonia.

## Figures and Tables

**Figure 1 animals-12-01204-f001:**
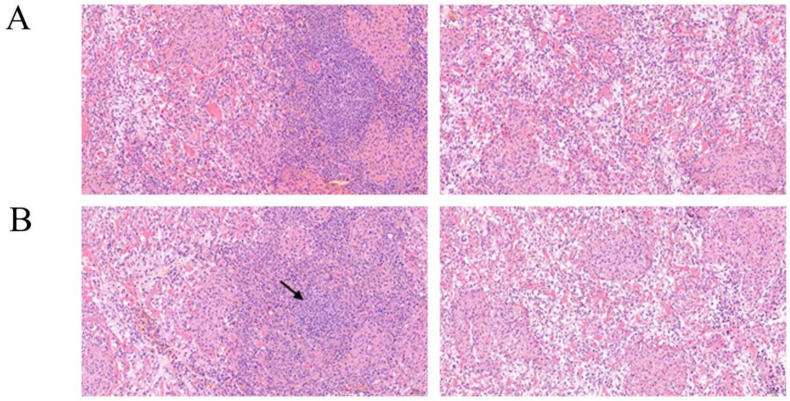
Histological structure of the spleens stained by hematoxylin and eosin (×20) (*n* = 3/group): (**A**) control group; (**B**) ammonia group. The histopathological lesions included the volume of splenic nodules decreased. Lymphocyte count (black arrow) was reduced.

**Figure 2 animals-12-01204-f002:**
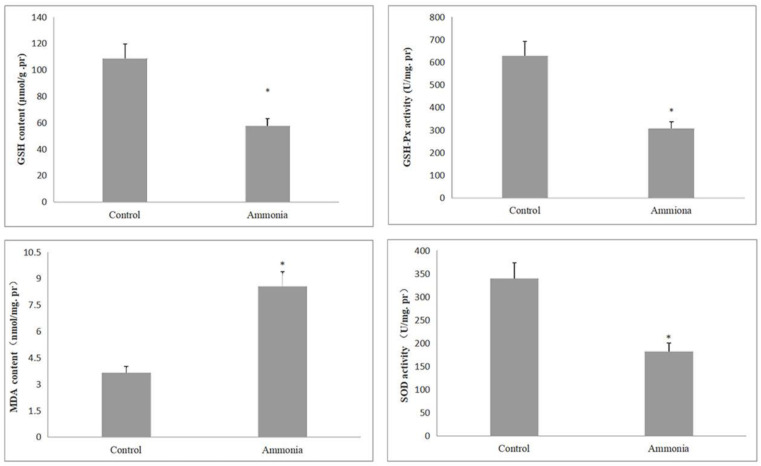
Changes in oxidative stress indicators induced by ammonia exposure (GSH-Px activity, SOD activity, GSH activity, and MDA content) in pig spleen. Data are expressed as mean ± standard deviation. * indicates a significant difference (*p* < 0.05).

**Figure 3 animals-12-01204-f003:**
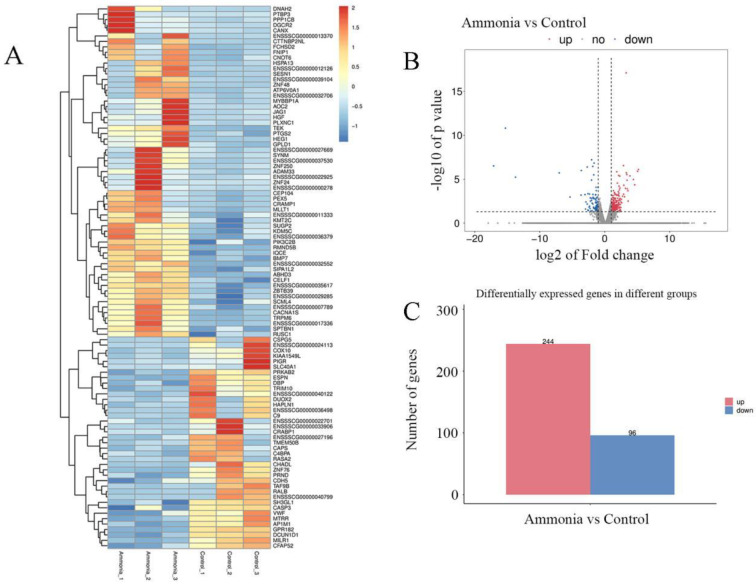
(**A**) Heat map of DEGs. The abscissa represents the sample, and the ordinate represents the gene. Different colors indicate different levels of gene expression. Red represents highly expressed genes, and dark blue represents low expressed genes. (**B**) Volcano plots of DEGs. The x-coordinate is log2 (fold-change), and the y-coordinate is −log10 (*p*-value). Red and blue dots represent upregulated and downregulated DEGs, respectively. (**C**) Column diagram of DEGs. The red column represents the number of upregulated genes, and the blue column represents the number of downregulated genes.

**Figure 4 animals-12-01204-f004:**
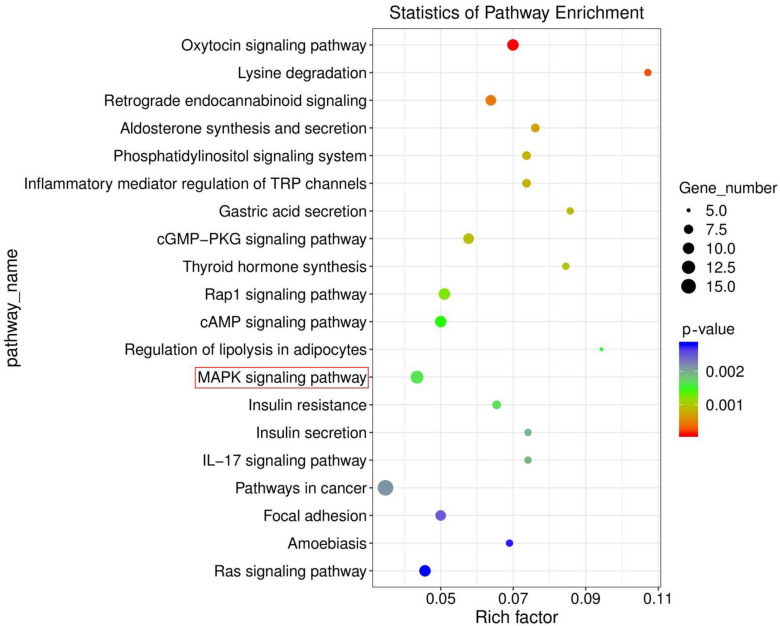
Top 20 pathways showed significant changes in histone methylation-related genes. MAPK signaling pathway was highlighted in red circles.

**Figure 5 animals-12-01204-f005:**
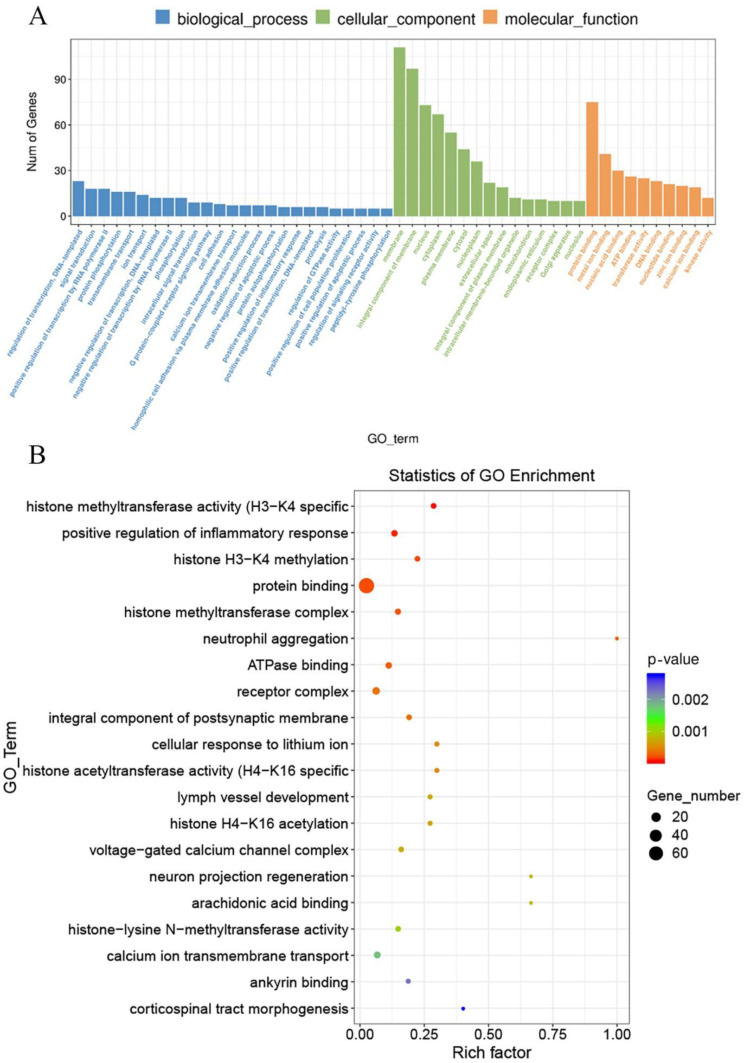
(**A**) GO enrichment analysis of DEGs. (**B**) Top 20 GO terms in GO enrichment analysis. Rich factor represents the number of DEGs enriched in the GO terms.

**Figure 6 animals-12-01204-f006:**
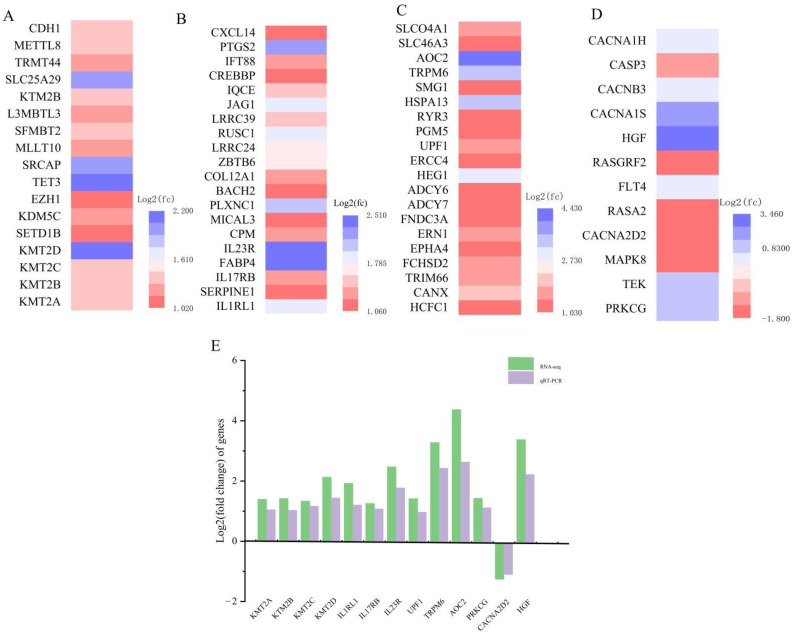
Forecast tendency of histone methyltransferase genes, oxidative stress, inflammatory, and MAPK signaling pathways in the pig spleen (*n* = 6/group): (**A**) histone methyltransferase genes; (**B**) oxidative stress; (**C**) inflammatory; (**D**) MAPK signaling pathways. (**E**) Comparison and analysis of RNA-Seq and qRT-PCR results.

**Table 1 animals-12-01204-t001:** Primers used for qRT-PCR.

Gene	Accession No.	Primer Sequence (5′-3′)
KMT2A	XM_021062911.1	F: CCTCTCGCTGCTTCACTTCA
		R: TGAGTTTCGGTCAGAGCCAC
KMT2B	XM_003127067.5	F: GAGCAAGATGATGCAGTGCG
		R: CACGGACTTGTAGTGGCCTT
KMT2C	XM_021079123.1	F: TTCGGATATAACTGCCCCGC
		R: GAGCAGAGAGAGCTGCTGTT
KMT2D	XM_021091593.1	F: CGATAGCTCTCCCAGCAAGG
		R: GTACGGGGCGTGACAGATAG
IL1RL1	XM_013995915.2	F: CAGGGAAGAAGCCACATCGT
		R: CAAAGCAAGCAGAGCACGTT
IL17RB	XM_005669645.3	F: ATCTGTGTGACGGGCAAGAG
		R: TTCTTTCATGCCTCCGGGTC
IL23R	NM_001137621.1	F: GGAAATCATCGGCCTTGCAG
		R: TTTGTGCTTTGCAATGAGGGA
UPF1	XM_021083471.1	F: GCCAGTTGTTGGCTGAGTTG
		R: GAGTCGCATGTCAGAGTCAGT
TRPM6	XM_021064975.1	F: CCAGCCACATAGGGCTTTGA
		R: GGATGACTGACCTCCCCTCT
AOC2	XM_003131393.4	F: AATGTTGGGGGTAGTGCCTG
		R: CACATCTGGGCGGACTCATT
PRKCG	XM_021094903.1	F: CCATTGGATCCCAGCACGAAT
		R: CTGCAGTTGTCAGCATCAGC
CACNA2D2	XM_021068821.1	F: ATGGACCAACGTGTACGAGG
		R: AGCAGGAACTCAAAATACTTGACC
HGF	XM_005667687.3	F: GCTGCTTCCCCTTCCTCTTT
		R: GCAAGAATTTGTGCCGGTGT

## Data Availability

The data presented in this study are available on reasonable request from the corresponding authors.

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
