# Peer review of "Transcriptome Revealed Exposure to the Environmental Ammonia Induced Oxidative Stress and Inflammatory Injury in Spleen of Fattening Pigs"

_animals, 2022, doi:10.3390/ani12091204_

Round 1

Reviewer 1 Report

This review manuscript "Transcriptome revealed exposure to the environmental ammonia induced oxidative stress and inflammatory injury in spleen of fatting pigs” is somewhat interesting, but the finding is superficial.

  1. Figure 1 had better show low power image to give an overview of the spleen structure and quantification if possible. In addition, how many mice have been observed per group should be shown.
  2. Lines 29,31 are using the term fattening pigs but some lines use fatting pig, what is a fatting pig? Could the author describe it in the manuscript?
  3. The results writing structure is not good. They are too simple. Authors should expand with why they do this and what they found.
  4. The figure legends are too simple, should describe better.
  5. the font size in Figure 3A is too small.

Reviewer 2 Report

This is an interesting study which aims to investigate the mechanisms of ammonia toxicity in cross-breeding fatting pigs as an experimental model. The results showed that 30 days-ammonia exposure at dose of 75.4 - 76.5 mg/m3 induces oxidative stress, morphological changes and inflammatory response through activation of MAPK signaling pathway in spleen tissues. Authors concluded that the observed differences in histone methylase gene expression may be the epigenetic mechanism of ammonia poisoning to the spleen. Since ammonia is among the most abundant and harmful gas in livestock production and the toxic mechanism is still not completely understood, this study worth publishing. For the benefits of readers, several points should be addressed before.

  • Please, define abbreviations upon first use.
  • It would be more useful for readers if the materials and methods were described in a little more detail.
  • Materials and methods, line 99: Please, explain the reason of the chosen ammonia doses (75.4 - 76.5 mg/m3).
  • Materials and methods, line 104 and section 2.5: Whether the samples were transferred into DNase/RNase free cryopreservation tubes for transcriptome analysis?
  • Line 115, “substance dismutase”: do you mean “superoxide dismutase”?
  • Line 113-120: Please, briefly describe the methods used (add IDs of the used kits). Line 120: Please, add “(Nanjing, China)”.
  • Line 130: Please, add (DEG)
  • Line 157 and 159: It should be “MDA” instead of “MAD”.
  • Figure 2, y-axis for MDA: “MDA levels” should be instead of “MDA activity”
  • Figures 3B and 3C: please add space (“AmmoniaVSControl” should be “Ammonia vs Control”, “-log10 of pvalue” should be “-log10 of p value”
  • Figure 6: Can the figure quality be improved?
  • Line 197/198 and 226: “Many studies” was written, and only one was cited. Please, add more references. For example, Line 198, authors can add reference “Xia C, Zhang X, Zhang Y, Li J, Xing H. Ammonia exposure causes the disruption of the solute carrier family gene network in pigs. Ecotoxicol Environ Saf. 2021 Mar 1;210:111870. doi: 10.1016/j.ecoenv.2020.111870. Epub 2021 Jan 10. PMID: 33440271.”
  • Line 215: one “and” should be deleted.

Reviewer 3 Report

The manuscript fits the scope of the journal and the overall research is well done. However, there are some issues that must be improved prior to acceptance.

1- The major criticism concerns the English language and style. There are many truncated sentences in the manuscript, with no verb and/or subject. I have highlighted several of them in the uploaded file. The discussion is difficult to read because it consists of short sentence.

2- The description of the methods is weak: for example, it is not mentioned whether the total RNA used for the qRT-PCR analysis has been cleaned from possible genomic DNA contamination; the description of the methodology used for qRT-PCR is completely missing.

Round 2

Reviewer 1 Report

The authors made a great improvement in the quality of the manuscript.